# Variational Bayesian Imaging with an Efficient Surrogate Score-based Prior

**Berthy T. Feng**                                                                                  *bfeng@caltech.edu*
*Computing and Mathematical Sciences*
*California Institute of Technology*

**Katherine L. Bouman**
*Computing and Mathematical Sciences*
*California Institute of Technology*

**Reviewed on OpenReview:** *https://openreview.net/forum?id=db2pFKVcm1*

## Abstract

We propose a surrogate function for efficient yet principled use of score-based priors in Bayesian imaging. We consider ill-posed inverse imaging problems in which one aims for a clean image posterior given incomplete or noisy measurements. Since the measurements do not uniquely determine a true image, a prior is needed to constrain the solution space. Recent work turned score-based diffusion models into principled priors for solving ill-posed imaging problems by appealing to an ODE-based log-probability function. However, evaluating the ODE is computationally inefficient and inhibits posterior estimation of high-dimensional images. Our proposed surrogate prior is based on the evidence lower bound of a score-based diffusion model. We demonstrate the surrogate prior on variational inference for efficient approximate posterior sampling of large images. Compared to the exact prior in previous work, our surrogate accelerates optimization of the variational image distribution by at least two orders of magnitude. We also find that our principled approach gives more accurate posterior estimation than non-variational diffusion-based approaches that involve hyperparameter-tuning at inference. Our work establishes a practical path forward for using score-based diffusion models as general-purpose image priors.

## 1 Introduction

Ill-posed image reconstruction arises when measurements of an object of interest are incomplete or noisy, making it impossible to uniquely determine the true image. Imaging in this setting requires a prior to constrain images according to desired statistics. From a Bayesian perspective, the prior influences both the uncertainty and the richness of the estimated image. Although diffusion-based generative models represent rich image priors, leveraging these priors for Bayesian image reconstruction remains a challenge. True posterior sampling with an unconditional diffusion model is intractable, so most previous methods either heavily approximate the posterior (Chung et al., 2023; Jalal et al., 2021; Kawar et al., 2022; Song et al., 2023a) or perform non-Bayesian conditional sampling (Choi et al., 2021; Chung et al., 2022a;b; Chung & Ye, 2022; Graikos et al., 2022; Song et al., 2022; Adam et al., 2022).

Recent work demonstrated how to turn score-based diffusion models into probabilistic priors (*score-based priors*) for variational Bayesian imaging (Feng et al., 2023). This method requires the exact probability of a proposed image to be evaluated with a computationally-expensive ordinary differential equation (ODE), requiring days to a week to reconstruct even a $32 \times 32$ image. Even so, this approach is appealing because it offers the same theoretical guarantees as traditional variational inference. This can sometimes be worth the computational cost for imaging applications in which measurements are expensive or difficult to obtain (e.g., black-hole imaging (EHTC, 2019), cryo-electron microscopy (Egelman, 2016; Levy et al., 2022), and

X-ray crystallography (Woolfson, 1997; Miao et al., 2008)). Still, it would be beneficial to have an efficient "surrogate" for this approach that can be used in the development and testing stages or simply to reduce computational costs. We present a method for variational Bayesian inference that is both principled and computationally efficient thanks to a surrogate score-based prior.

Computing exact probabilities under a diffusion model is inefficient or even intractable, but computing the evidence lower bound (ELBO) (Song et al., 2021a; Ho et al., 2020) is computationally efficient and feasible for high-dimensional images. We propose to use this lower bound as a surrogate for the exact score-based prior. In particular, we use the ELBO function as a substitute for the exact log-probability function, and it can be plugged into any inference algorithm that requires the value or gradient of the posterior log-density. For variational inference of an image posterior, we find at least two orders of magnitude in speedup of optimizing the variational distribution. Furthermore, our approach reduces GPU memory requirements, as there is no need to evaluate and backpropagate through an ODE. These efficiency improvements make it practical to perform principled inference with score-based priors.

In this paper, we describe our variational-inference approach to estimate a posterior with a surrogate score-based prior.[1] We provide experimental results to validate the proposed surrogate, including high-dimensional posterior samples of sizes up to $256 \times 256$. In the setting of accelerated MRI, we quantify time- and memory-efficiency improvements of the surrogate over the exact prior. We also demonstrate that our approach achieves more accurate posterior estimation and higher-fidelity image reconstructions than diffusion-based methods that deviate from true Bayesian inference. Finally, we demonstrate how our approach can be used for black-hole imaging, in which images must be carefully reconstructed from sparse telescope measurements.

## 2 Related work

### 2.1 Bayesian inverse imaging

Imaging can be framed as an inverse problem in which a hidden image $\mathbf{x}^* \in \mathbb{R}^D$ must be recovered from measurements $\mathbf{y} \in \mathbb{R}^M$, where

$$\mathbf{y} = f(\mathbf{x}^*) + \mathbf{n}.$$

Usually the forward model $f : \mathbb{R}^D \to \mathbb{R}^M$ and statistics of the noise $\mathbf{n} \in \mathbb{R}^M$ are assumed to be known.

Bayesian imaging accounts for the uncertainty added by the measurement process by formulating a posterior distribution $p(\mathbf{x} \mid \mathbf{y})$, whose log-density can be decomposed into a likelihood term and a prior term:

$$\log p(\mathbf{x} \mid \mathbf{y}) = \log p(\mathbf{y} \mid \mathbf{x}) + \log p(\mathbf{x}) + \text{const.} \tag{1}$$

Given a log-likelihood function $\log p(\mathbf{y} \mid \mathbf{x})$ and a prior log-probability function $\log p(\mathbf{x})$, we can use established techniques for sampling from the posterior, such as Markov chain Monte Carlo (MCMC) (Brooks et al., 2011) or variational inference (VI) (Blei et al., 2017). MCMC algorithms generate a Markov chain whose stationary distribution is the posterior, but they are generally slow to converge for high-dimensional data like images. VI instead approximates the posterior with a tractable distribution. The variational distribution is usually parameterized and thus can be efficiently optimized to represent high-dimensional data distributions. Deep Probabilistic Imaging (DPI) (Sun & Bouman, 2021; Sun et al., 2022) proposed an efficient VI approach specifically for computational imaging with image regularizers. In DPI, the variational distribution is a discrete normalizing flow (Kobyzev et al., 2020), which is an invertible generative model capable of representing complex distributions.

The log-prior term in Eq. 1 is often defined through a regularizer. Traditional imaging approaches use hand-crafted regularizers including total variation (TV) (Vogel & Oman, 1996), maximum entropy (Gull & Skilling, 1984; Narayan & Nityananda, 1986), and the L1 norm (Candes & Romberg, 2007) or data-driven regularizers such as through Gaussian mixture models (Zoran & Weiss, 2011; 2012). Various deep-learned regularizers have been proposed, including adversarial regularizers (Lunz et al., 2018) and input-convex neural networks (Mukherjee et al., 2020; Shumaylov et al., 2023), but they impose training or architecture restrictions that may limit generalizability and expressiveness.

---

[1]Code is available at `https://github.com/berthyf96/score_prior`.

We note that this perspective of Bayesian imaging with standalone priors differs from that of other learning-based approaches to inverse problems. Many approaches train an end-to-end neural network on a paired dataset of measurements and ground-truth images (Pathak et al., 2016; Iizuka et al., 2017; Yu et al., 2019; Zhang et al., 2022; 2020; Yin et al., 2021; Saharia et al., 2022; Delbracio et al., 2021; Delbracio & Milanfar, 2023). However, for every new measurement distribution, a new paired dataset would have to be obtained, and the network would have to be re-trained. This can become cumbersome when one wishes to analyze different measurement settings with the same prior (e.g., when exploring different MRI acquisition strategies). It also precludes analyzing the effects of different priors (e.g., by imposing different assumptions in imaging a black hole (EHTC, 2019)) since the implicit prior of the network changes every time it is re-trained. Our approach, on the other hand, only requires a dataset of clean images for the prior, and the same learned prior can be used across different inverse problems by being paired with different measurement-likelihood functions in a modular way. Large-scale datasets of clean images exist for many applications, such as the fastMRI (Zbontar et al., 2018) dataset for accelerated MRI. It is also usually possible to create a clean dataset through costly acquisition of ideal images or simulation.

## 2.2 Diffusion models for inverse problems

Diffusion models (Sohl-Dickstein et al., 2015; Ho et al., 2020; Song & Ermon, 2019; Song et al., 2019; 2021b) learn to model a rich image distribution that could be useful as a prior for image reconstruction. A diffusion model generates an image by starting from an image of noise and gradually denoising it until it becomes a clean image. We discuss this process, known as *reverse diffusion*, in more detail in Sec. 3.1.

Given an inverse problem, simply adapting a trained diffusion model to sample from the posterior instead of the learned prior is intractable (Feng et al., 2023). Therefore, most diffusion-based approaches do not infer a true posterior. Some project images onto a measurement-consistent subspace (Song et al., 2022; Chung et al., 2022b; Chung & Ye, 2022; Choi et al., 2021; Chung et al., 2022a), but the projection does not account for measurement noise and might pull images away from the true posterior. Others follow a gradient toward higher measurement-likelihood throughout reverse diffusion (Chung et al., 2023; Jalal et al., 2021; Graikos et al., 2022; Kawar et al., 2022; Adam et al., 2022; Song et al., 2023a; Mardani et al., 2023), but they heavily approximate the posterior. Overall, these diffusion-based methods require hyperparameter-tuning of the measurement weight. As soon as hyperparameters are introduced, there is no guarantee of sampling from a posterior that represents the true uncertainty. These diffusion-based methods are not principled enough for scientific and medical applications that require accurate uncertainty quantification.

**Score-based priors** Recent work proposed an alternative perspective: turning a score-based diffusion model into a standalone, probabilistic prior (*score-based prior*) that can be paired with any measurement-likelihood function and plugged into established variational-inference approaches. Rigorous variational inference is usually more computationally intensive than the aforementioned diffusion-based approaches, so this type of approach is best-suited for applications in which finding the best posterior for a given set of measurements is worth the computational overhead.

However, the approach of Feng et al. (2023) is too memory-intensive to feasibly handle large images. Even for small images, it is too time-intensive to enable fast development. This is due to a log-density function based on an ODE (see Sec. 3.2) that is computationally expensive. Our work proposes an efficient surrogate that makes this type of principled approach practical for high-dimensional inference. While other efforts have been made to speed up diffusion-model sampling, such as by reducing the number of function evaluations (Karras et al., 2022; Zhang & Chen, 2022; Song et al., 2023b), performing latent diffusion (Rombach et al., 2022; Avrahami et al., 2023), or learning more efficient reverse processes (Lipman et al., 2022; De Bortoli et al., 2021), these approaches either break the tractability of image probabilities or only incrementally improve speed. We aim to fundamentally improve efficiency without altering the diffusion model.

# 3 Background

In this section, we review background on score-based diffusion models with an emphasis on evaluating probabilities of images under a trained diffusion model. We then describe how a diffusion process gives rise to an efficient denoising-based lower bound on these image probabilities.

## 3.1 Score-based diffusion models

The core idea of a diffusion model is that it transforms a simple distribution $\pi$ to a complex image distribution through a gradual process. We follow the popular framework of denoising diffusion models, which transform noise samples from $\pi = \mathcal{N}(\mathbf{0}, \mathbf{I})$ to clean samples from the data distribution $p_{\text{data}}$ through gradual denoising. With knowledge of the noise distribution and denoising process, we can assess the probability of a novel image under this generative model.

The transformation from a simple distribution to a complex one occurs over many steps. To determine how the data distribution should look at each step of denoising, we turn to a stochastic differential equation (SDE) that describes a diffusion process from clean images to noise (Song et al., 2021b). The diffusion SDE is defined on the time interval $t \in [0, T]$ as

$$\mathrm{d}\mathbf{x} = \mathbf{f}(\mathbf{x}, t)\mathrm{d}t + g(t)\mathrm{d}\mathbf{w}, \tag{2}$$

where $\mathbf{w} \in \mathbb{R}^D$ denotes Brownian motion; $g(t) \in \mathbb{R}$ is the diffusion coefficient, which controls the rate of noise increase; and $\mathbf{f}(\cdot, t) : \mathbb{R}^D \to \mathbb{R}^D$ is the drift coefficient, which controls the deterministic evolution of $\mathbf{x}(t)$. By defining a stochastic trajectory $\{\mathbf{x}(t)\}_{t \in [0,T]}$, this SDE gives rise to a time-dependent probability distribution $p_t$, which is the marginal distribution of $\mathbf{x}(t)$. We construct $\mathbf{f}(\cdot, t)$ and $g(t)$ so that if $p_0 = p_{\text{data}}$, then $p_T \approx \pi$. Image generation amounts to reversing the diffusion, which requires the gradient of the data log-density (*score*) at every noise level in order to nudge images toward high probability under $p_{\text{data}}$. A convolutional neural network $\mathbf{s}_\theta$ known as a *score model* learns to approximate the true score: $\mathbf{s}_\theta(\mathbf{x}, t) \approx \nabla_{\mathbf{x}} \log p_t(\mathbf{x})$.

**Sampling with a reverse-time SDE** Once trained, $\mathbf{s}_\theta(\mathbf{x}, t)$ is used to reverse diffusion and generate clean images from noise. This results in a distribution $p_\theta^{\text{SDE}}$, denoted as such because it is determined by a reverse-time SDE (Song et al., 2021b):

$$\mathrm{d}\mathbf{x} = \left[\mathbf{f}(\mathbf{x}, t) - g(t)^2 \mathbf{s}_\theta(\mathbf{x}, t)\right] \mathrm{d}t + g(t)\mathrm{d}\bar{\mathbf{w}}, \tag{3}$$

where $\bar{\mathbf{w}} \in \mathbb{R}^D$ denotes Brownian motion, and $\mathbf{f}(\cdot, t)$ and $g(t)$ are the same as in Eq. 2. To generate an image, we first sample $\mathbf{x}(T) \sim \mathcal{N}(\mathbf{0}, \mathbf{I})$ and then numerically solve the reverse-time SDE for $\mathbf{x}(0)$. The marginal distribution of $\mathbf{x}(0)$ is denoted by $p_\theta^{\text{SDE}}$, which is close to $p_{\text{data}}$ when the score model is well-trained. Intuitively, Eq. 3 explains how to denoise an image $\mathbf{x}(t)$ using the score model $\mathbf{s}_\theta$ in order to get closer to the clean image distribution. Indeed, Eq. 3 is exactly the reverse of the forward-time SDE (Eq. 2) (Anderson, 1982) when replacing $\nabla_{\mathbf{x}} \log p_t(\mathbf{x})$ with $\mathbf{s}_\theta(\mathbf{x}, t)$, which means that removing noise corresponds to reaching higher probability under the data distribution.

## 3.2 Image probabilities

The generated image distribution $p_\theta^{\text{SDE}}$ theoretically assigns density to any image. However, reverse diffusion does not lead to an image distribution with tractable probabilities. In this subsection, we describe two workarounds: one based on an ODE and the other based on an ELBO related to denoising score-matching.

To compute the probability of an image $\mathbf{x}$ under $p_\theta^{\text{SDE}}$, we need to invert it from $\mathbf{x}(0) = \mathbf{x}$ to $\mathbf{x}(T)$. However, this is not tractable through the SDE: just as it is intractable to reverse a random walk, it is intractable to account for all the possible starting points $\mathbf{x}(T)$ that could have resulted in $\mathbf{x}(0)$ through the stochastic process. Probability computation calls for an invertible process that lets us map any point from $p_{\text{data}}$ to a point from $\mathcal{N}(\mathbf{0}, \mathbf{I})$ and vice versa.

**Probability flow ODE** The *probability flow ODE* (Song et al., 2021b) defines an invertible sampling function for a distribution $p_\theta^{\text{ODE}}$ theoretically the same as $p_\theta^{\text{SDE}}$. It is given by

$$\frac{d\mathbf{x}}{dt} = \mathbf{f}(\mathbf{x}, t) - \frac{1}{2}g(t)^2 \mathbf{s}_\theta(\mathbf{x}, t) =: \tilde{\mathbf{f}}_\theta(\mathbf{x}, t). \tag{4}$$

The absence of Brownian motion makes it possible to solve this ODE in both directions of time. To compute the log-probability of an image $\mathbf{x}$, we map $\mathbf{x}(0) = \mathbf{x}$ to its corresponding noise image $\mathbf{x}(T)$. Under the framework of neural ODEs (Chen et al., 2018), the log-probability is given by the log-probability of $\mathbf{x}(T)$ under $\mathcal{N}(\mathbf{0}, \mathbf{I})$ plus a normalizing factor accounting for the change in density through time:

$$\log p_\theta^{\text{ODE}}(\mathbf{x}(0)) = \log \pi(\mathbf{x}(T)) + \int_0^T \nabla \cdot \tilde{\mathbf{f}}_\theta(\mathbf{x}(t), t) dt \tag{5}$$

for $\mathbf{x}(0) = \mathbf{x}$. Although tractable to evaluate with an ODE solver, this log-probability function is computationally expensive, requiring hundreds to thousands of discrete ODE time steps to accurately evaluate. Additional time and memory costs are incurred by backpropagation through the ODE and Hutchinson-Skilling trace estimation of the divergence.

**Equivalence of $p_\theta^{\text{SDE}}$ and $p_\theta^{\text{ODE}}$** Song et al. (2021a) proved that if $\mathbf{s}_\theta(\mathbf{x}, t) \equiv \nabla_\mathbf{x} \log p_t(\mathbf{x}, t)$ for all $t \in [0, T]$ and $p_T = \pi$, then $p_\theta^{\text{ODE}} = p_\theta^{\text{SDE}} = p_{\text{data}}$. In our work, we assume that $\mathbf{s}_\theta(\mathbf{x}, t) \approx \nabla_\mathbf{x} \log p_t(\mathbf{x}, t)$ for almost all $\mathbf{x} \in \mathbb{R}^D$ and $t \in [0, T]$ and that $p_T \approx \mathcal{N}(\mathbf{0}, \mathbf{I})$, so that $p_\theta^{\text{ODE}} \approx p_\theta^{\text{SDE}} \approx p_{\text{data}}$. This assumption empirically performed well in previous work that appealed to $p_\theta^{\text{ODE}}$ as the exact probability distribution of the diffusion model (Feng et al., 2023; Song et al., 2021b).

### 3.2.1 Evidence lower bound

In lieu of an exact log-probability function, Song et al. (2021a) derived an evidence lower bound $b_\theta^{\text{SDE}}$ for $p_\theta^{\text{SDE}}$ such that $b_\theta^{\text{SDE}}(\mathbf{x}) \leq \log p_\theta^{\text{SDE}}(\mathbf{x})$ for any proposed image $\mathbf{x}$. Essentially, this lower bound corresponds to how well the diffusion model is able to denoise a given image: an image with high probability under the diffusion model is easy to denoise, whereas a low-probability image is difficult.

The lower bound, or the negative "denoising score-matching loss" (Song et al., 2021a), is defined as

$$b_\theta^{\text{SDE}}(\mathbf{x}) := \mathbb{E}_{p_{0T}(\mathbf{x}'|\mathbf{x})} \left[ \log \pi(\mathbf{x}') \right] - \frac{1}{2} \int_0^T g(t)^2 h(t) dt, \tag{6}$$

where

$$h(t) := \mathbb{E}_{p_{0t}(\mathbf{x}'|\mathbf{x})} \left[ \|\mathbf{s}_\theta(\mathbf{x}', t) - \nabla_{\mathbf{x}'} \log p_{0t}(\mathbf{x}' \mid \mathbf{x})\|_2^2 - \|\nabla_{\mathbf{x}'} \log p_{0t}(\mathbf{x}' \mid \mathbf{x})\|_2^2 - \frac{2}{g(t)^2} \nabla_{\mathbf{x}'} \cdot \mathbf{f}(\mathbf{x}', t) \right]. \tag{7}$$

$p_{0t}(\mathbf{x}' \mid \mathbf{x})$ denotes the transition distribution from $\mathbf{x}(0) = \mathbf{x}$ to $\mathbf{x}(t) = \mathbf{x}'$. For a drift coefficient that is linear in $\mathbf{x}$, this transition distribution is Gaussian: $p_{0t}(\mathbf{x}' \mid \mathbf{x}) = \mathcal{N}(\mathbf{x}'; \alpha(t)\mathbf{x}, \beta(t)^2 \mathbf{I})$. This means that the gradient $\nabla_{\mathbf{x}'} \log p_{0t}(\mathbf{x}' \mid \mathbf{x})$ is directly proportional to the Gaussian noise that is subtracted from $\mathbf{x}'$ to get $\mathbf{x}$. Eq. 6 is efficient to compute since we can evaluate it by adding Gaussian noise to $\mathbf{x}$ without having to solve an initial-value problem as with the ODE. In fact, Eq. 6 is closely related to the denoising score-matching objective used to train diffusion models (Song et al., 2021b).

Intuitively, we can interpret Eq. 6 as associating an image's probability with how well the score model $\mathbf{s}_\theta$ could denoise that image if it underwent diffusion. This is represented by the first term in $h(t)$ (Eq. 7). To assess the probability of an image $\mathbf{x}$, we perturb it with Gaussian noise to get $\mathbf{x}'$ and then ask the score model to estimate the noise that was added. If $\mathbf{s}_\theta(\mathbf{x}, t)$ accurately estimates the noise, then $\|\mathbf{s}_\theta(\mathbf{x}', t) - \nabla_{\mathbf{x}'} \log p_{0t}(\mathbf{x}' \mid \mathbf{x})\|_2^2$ is small, and the value of $b_\theta^{\text{SDE}}(\mathbf{x})$ becomes larger. The remaining terms in $h(t)$ are normalizing factors independent of $\theta$, and $\mathbb{E}_{p_{0T}(\mathbf{x}'|\mathbf{x})} \left[ \log \pi(\mathbf{x}') \right]$ accounts for the probabilities of the noise images $\mathbf{x}(T)$ that could result from $\mathbf{x}$ being entirely diffused.

## 4  Method

Inspired by previous theoretical work (Song et al., 2021a), we apply $b_\theta^{\text{SDE}}$ as an efficient surrogate for the exact score-based prior in Bayesian imaging. In this section, we describe our approach for efficient inference of an approximate posterior given a score-based prior.

### 4.1  Approximating the posterior with VI

Given measurements $\mathbf{y} \in \mathbb{R}^M$ (with a known log-likelihood function) and a score-based diffusion model with parameters $\theta$ as the prior, our goal is to sample from the image posterior $p_\theta(\mathbf{x} \mid \mathbf{y})$. Following VI, we optimize the parameters of a variational distribution to approximate $p_\theta(\mathbf{x} \mid \mathbf{y})$.

Let $q_\phi$ denote the variational distribution with parameters $\phi$. We would like to optimize $\phi$ to minimize the KL divergence from $q_\phi$ to the target posterior:

$$\phi^* = \arg\min_\phi D_{\text{KL}}(q_\phi \| p_\theta(\cdot \mid \mathbf{y})) \tag{8}$$

$$= \arg\min_\phi \mathbb{E}_{\mathbf{x} \sim q_\phi} \left[ -\log p(\mathbf{y} \mid \mathbf{x}) - \log p_\theta^{\text{ODE}}(\mathbf{x}) + \log q_\phi(\mathbf{x}) \right]. \tag{9}$$

$q_\phi$ can be any parameterized tractable distribution. It could be a Gaussian distribution with a diagonal covariance so that $\phi := [\mu^\top, \sigma^\top]^\top$, where $\mu \in \mathbb{R}^D$ and $\sigma \in \mathbb{R}^D$ ($\sigma > \mathbf{0}$) are the mean and pixel-wise standard deviation. $q_\phi$ could also be a RealNVP normalizing flow as it is in DPI (Sun & Bouman, 2021).

To circumvent the computational challenges of evaluating the prior term $\log p_\theta^{\text{ODE}}(\mathbf{x})$, we replace it with the surrogate $b_\theta^{\text{SDE}}(\mathbf{x})$. This results in the following objective:

$$\phi^* = \arg\min_\phi \mathbb{E}_{\mathbf{x} \sim q_\phi} \left[ -\log p(\mathbf{y} \mid \mathbf{x}) - b_\theta^{\text{SDE}}(\mathbf{x}) + \log q_\phi(\mathbf{x}) \right]. \tag{10}$$

We can also think of $b_\theta^{\text{SDE}}$ as replacing the intractable $\log p_\theta^{\text{SDE}}$ in Eq. 8. Since $-\log p_\theta^{\text{SDE}} \leq -b_\theta^{\text{SDE}}$, *our surrogate objective minimizes the upper bound of the KL divergence involving $p_\theta^{SDE}$.*

### 4.2  Implementation details

**Evaluating $b_\theta^{\mathbf{SDE}}(\mathbf{x})$**   The formula for $b_\theta^{\text{SDE}}(\mathbf{x})$ (Eq. 6) contains a time integral and expectation over $p_{0t}(\mathbf{x}' \mid \mathbf{x})$ that can be estimated with numerical methods. Following Song et al. (2021a), we use importance sampling with time samples $t \sim p(t)$ for the time integral and Monte-Carlo approximation with noisy images $\mathbf{x}' \sim \mathcal{N}(\alpha(t)\mathbf{x}, \beta(t)^2\mathbf{I})$ for the expectation. The proposal distribution $p(t) := \frac{g(t)^2}{\beta(t)^2 Z}$ was empirically verified to result in lower variance in the estimation of $b_\theta^{\text{SDE}}(\mathbf{x})$ (Song et al., 2021a). We provide the following formula used in our implementation, which estimates the time integral with importance sampling and the expectation with Monte-Carlo approximation, for reference:

$$b_\theta^{\text{SDE}}(\mathbf{x}) \approx \frac{1}{N_{\mathbf{z}}} \sum_{j=1}^{N_{\mathbf{z}}} \log \pi\left(\mathbf{x}'_j\right) - \frac{1}{2N_t N_{\mathbf{z}}} \sum_{i=1}^{N_t} Z\beta(t)^2 \sum_{j=1}^{N_{\mathbf{z}}} \left[ \left\| \mathbf{s}_\theta(\mathbf{x}'_{ij}, t_i) + \frac{\mathbf{z}_{ij}}{\beta(t_i)} \right\|_2^2 - \left\| \frac{\mathbf{z}_{ij}}{\beta(t_i)} \right\|_2^2 - \frac{2}{g(t_i)^2} \nabla_{\mathbf{x}'_{ij}} \cdot \mathbf{f}(\mathbf{x}'_{ij}, t_i) \right] \tag{11}$$

$$\text{s.t.} \quad t_i \sim p(t), \ \mathbf{z}_{ij} \sim \mathcal{N}(\mathbf{0}, \mathbf{I}), \ \mathbf{x}'_{ij} = \alpha(t_i)\mathbf{x} + \beta(t_i)\mathbf{z}_{ij},$$

$$\mathbf{x}'_j \sim \mathcal{N}(\alpha(T)\mathbf{x}, \beta(T)^2\mathbf{I}) \quad \forall i = 1, \ldots, N_t, j = 1, \ldots, N_{\mathbf{z}}.$$

$N_t$ time samples and $N_{\mathbf{z}}$ noise samples are taken to approximate the time integral and expectation over $p_{0t}(\mathbf{x}' \mid \mathbf{x})$, respectively. In our experiments, $N_t = N_{\mathbf{z}} = 1$. We find that increasing the number of time and noise samples does not efficiently decrease variance in the estimated value of $b_\theta^{\text{SDE}}(\mathbf{x})$.

**Optimization**   We use stochastic gradient descent to optimize $\phi$, Monte-Carlo-approximating the expectation in Eq. 10 with a batch of $\mathbf{x} \sim q_\phi$. Estimating $b_\theta^{\text{SDE}}(\mathbf{x})$ has higher variance than estimating $\log p_\theta^{\text{ODE}}(\mathbf{x})$. For instance, in Fig. 4, $b_\theta^{\text{SDE}}(\mathbf{x})$ with $N_t = 2048$, $N_{\mathbf{z}} = 1$ shows higher variance than $\log p_\theta^{\text{ODE}}(\mathbf{x})$ with 16 trace estimators. A lower learning rate can help mitigate training instabilities caused by variance. For example, in Fig. 3b the learning rate with the exact prior was 0.0002, while the learning rate with the surrogate prior was 0.00001. Please refer to Appendix B for more optimization details.

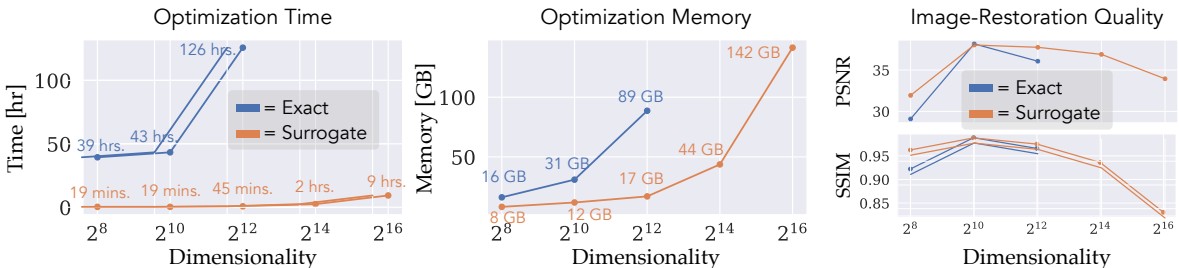

Figure 1: Computational efficiency of proposed surrogate prior vs. exact prior. For each image size, we estimated a posterior of images conditioned on 4×-accelerated MRI measurements of a knee image, using a Gaussian distribution with diagonal covariance as the variational distribution. The hardware was 4x NVIDIA RTX A6000. The surrogate prior allows for variational inference of image sizes that are prohibitively large for the exact prior. For image sizes supported by the exact prior, the surrogate improved total optimization time by over 120× while using less memory and scaling better with image size. "Image-Restoration Quality" verifies that optimization with the surrogate was done fairly, as the PSNR and SSIM of the converged posterior (averaged over 128 samples) are at least as high as with the exact prior.

## 5 Experiments

We validate our approach on the tasks of accelerated MRI, denoising, reconstruction from low spatial frequencies, and black-hole interferometric imaging. See Appendix A for details about the forward models.

### 5.1 Efficiency improvements

In Tab. 1 and Fig. 1, we quantify the efficiency improvements of the surrogate prior for an accelerated MRI task at different image resolutions. We drew a test image from the fastMRI knee dataset (Zbontar et al., 2018) and resized it to $16 \times 16$, $32 \times 32$, $64 \times 64$, $128 \times 128$, and $256 \times 256$. For each image size, we trained a score model on images of the corresponding size from the fastMRI training set of single-coil knee scans. We then optimized a Gaussian distribution with diagonal covariance to approximate the posterior. The batch size was 64 for the surrogate and 32 for the exact prior (a smaller batch size was needed to fit $64 \times 64$ optimization into GPU memory). Convergence was defined via a minimum acceptable change in the estimated posterior mean between optimization steps.

| Image size | Surrogate | Exact |
|---|---|---|
| $16 \times 16$ | 0.029 | 19.5 |
| $32 \times 32$ | 0.038 | 41.9 |
| $64 \times 64$ | 0.090 | 123 |
| $128 \times 128$ | 0.294 | N/A |
| $256 \times 256$ | 1.115 | N/A |

Table 1: Iteration time [sec/step]. A step of gradient-based optimization of the variational distribution is two to three orders of magnitude faster with the surrogate prior.

We find at least two orders of magnitude in time improvement with the surrogate prior. Tab. 1 compares the iteration time between the two priors. Fig. 1 compares the total time it takes to optimize the variational distribution. The surrogate also improves memory consumption, which in turn enables optimizing higher-dimensional posteriors. Following standard practice, we just-in-time (JIT) compile the optimization step to reduce time/step at the cost of GPU memory. Fig. 1 shows how the surrogate prior significantly reduces memory requirements and scales better with image size. The exact prior could only handle up to $64 \times 64$ before exceeding GPU memory (we tested on 4x 48GB GPUs). While memory could be reduced with a smaller batch size, this would make optimization more time-consuming. On the other hand, our surrogate prior supports much larger images, as we demonstrate in Fig. 2 for $256 \times 256^2$ MRI with a Gaussian-approximated posterior. This type of principled inference of high-dimensional image posteriors was not possible before with the exact score-based prior.

---

[2]Larger images may be feasible, but their larger memory footprint might restrict the possible batch size and complexity of the variational distribution.

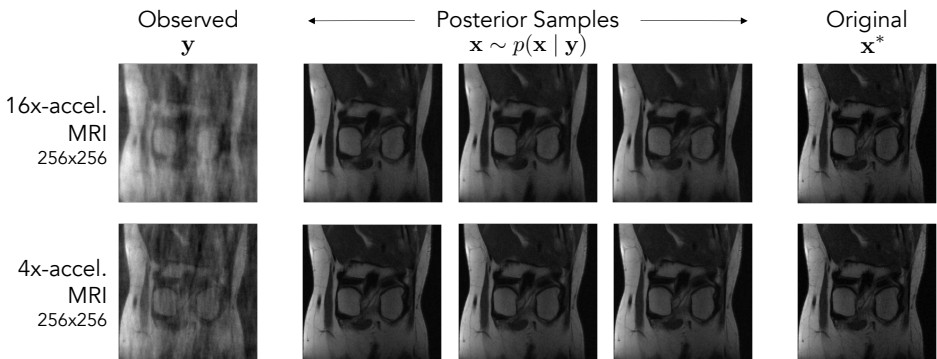

Figure 2: High-dimensional Bayesian inference with a surrogate score-based prior. Here we show posterior samples for accelerated MRI of $256\times256$ knee images, approximated via variational inference with a surrogate score-based prior. The first row shows reconstruction from $16\times$-reduced MRI measurements. The second row shows reconstruction given more $\kappa$-space measurements, i.e., $4\times$-reduced MRI. Bayesian imaging at this image resolution is computationally infeasible with the previous ODE-based approach (Feng et al., 2023). Our proposed surrogate enables efficient yet principled inference with diffusion-model priors, resulting in inferred posteriors where the true image is within three standard deviations of the posterior mean for 96% and 99% of the pixels for $16\times$- and $4\times$-acceleration, respectively.

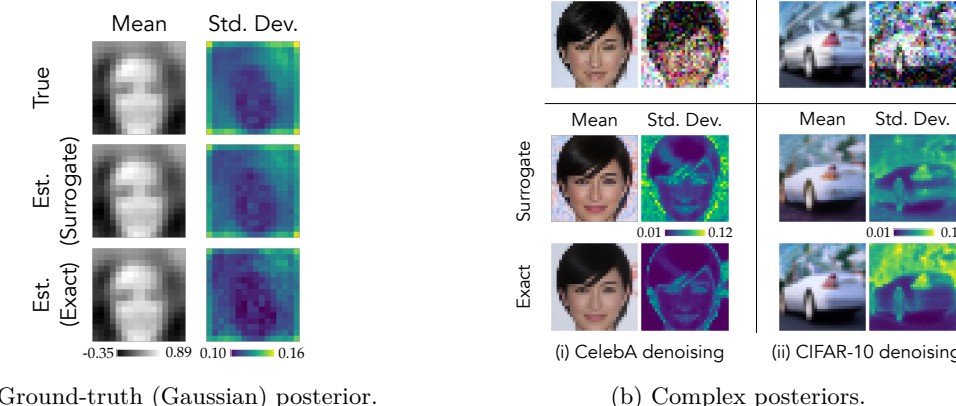

(a) Ground-truth (Gaussian) posterior.

(b) Complex posteriors.

Figure 3: Estimated posteriors under surrogate vs. exact prior. For each task, the variational distribution is a RealNVP, and the score model is the same between both prior functions. **(a)** Both prior functions help recover the correct (Gaussian) posterior. The score-based prior was trained on samples from a known Gaussian distribution (originally fit to $16 \times 16$ face images), and the measurements are the lowest 6.25% spatial frequencies of a test image from the prior. Since the prior and likelihood are both Gaussian, we know the ground-truth Gaussian posterior. **(b)** Estimated posteriors for (i) denoising a CelebA image and (ii) denoising a CIFAR-10 image. Std. dev. is averaged across the three color channels. The score-based prior was trained on CelebA in (i) and CIFAR-10 in (ii). Both prior functions result in comparable image quality; visual differences appear mostly in the image background.

## 5.2 Posterior estimation under surrogate vs. exact

The surrogate prior $b_\theta^{\mathrm{SDE}}$ may not be an identical substitute for the exact prior $\log p_\theta^{\mathrm{ODE}}$. Importantly, though, we verify in Fig. 3a that both the surrogate and exact prior recover a ground-truth Gaussian posterior derived from a Gaussian likelihood and prior. The variational distribution is a RealNVP. The score model (used by both the surrogate and exact prior) was trained on samples from the known prior.

Nonetheless, the surrogate could result in a different locally-optimal variational posterior, particularly if the posterior leads to many local minima in the variational objective. Fig. 3b compares posteriors (with unknown true distributions) approximated by a RealNVP under the surrogate versus exact prior. For each task (CelebA denoising and CIFAR-10 denoising), both prior functions used the same trained score model. We observe in these comparisons that most of the differences appear in the image background and that both priors result in a plausible mean reconstruction and uncertainty.

**Visualizing the bound gap throughout optimization** helps shed light on why the two priors converge to different solutions even with the same underlying score model. Fig. 4 shows probabilities of samples generated by $q_\phi$ (in this case a RealNVP) as optimization progresses. At each checkpoint of $q_\phi$, we plot $b_\theta^{\text{SDE}}(\mathbf{x})$ versus $\log p_\theta^{\text{ODE}}(\mathbf{x})$ for samples $\mathbf{x} \sim q_\phi$ coming from both the exact and surrogate optimization of $q_\phi$. We find that the surrogate is a valid bound for the ODE log-density: $b_\theta^{\text{SDE}}(\mathbf{x}) \leq \log p_\theta^{\text{ODE}}(\mathbf{x})$ for all $\mathbf{x} \sim q_\phi$, except for some outliers due to variance of $b_\theta^{\text{SDE}}(\mathbf{x})$. However, optimization follows a different trajectory depending on the prior. With the surrogate, samples $\mathbf{x} \sim q_\phi$ tend toward a region where the bound gap is small. Meanwhile, the exact prior follows a loss landscape whose structure appears to be independent of the lower bound. Note that samples from $q_\phi$ optimized under the exact prior obtain higher values of $b_\theta^{\text{SDE}}(\mathbf{x})$ than samples obtained under the surrogate. Fig. 4 suggests that gradients under the surrogate tend to push the $q_\phi$ distribution along the boundary of equality between $b_\theta^{\text{SDE}}$ and $\log p_\theta^{\text{ODE}}$. This constrains the path taken through gradient descent and subsequently the converged solution.

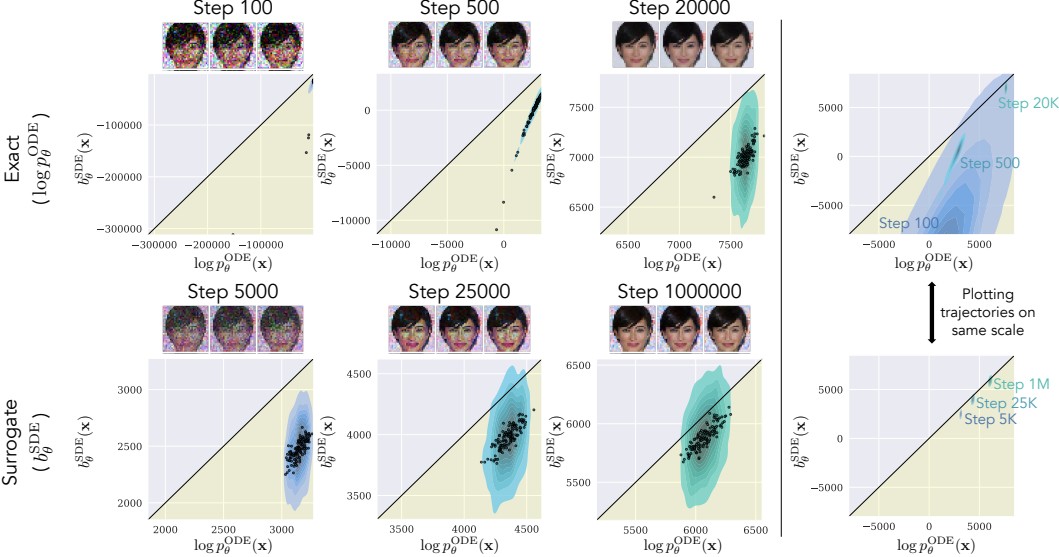

Figure 4: $b_\theta^{\text{SDE}}(\mathbf{x})$ vs. $\log p_\theta^{\text{ODE}}(\mathbf{x})$ for samples $\mathbf{x} \sim q_\phi$ as optimization of $\phi$ progresses. The task is from Fig. 3b(i). For each plot, we took 128 samples $\mathbf{x} \sim q_\phi$ and performed 20 estimates each of $\log p_\theta^{\text{ODE}}(\mathbf{x})$ and $b_\theta^{\text{SDE}}(\mathbf{x})$ (approximated with $N_t = 2048$ for reduced variance). The density map is a KDE plot of all $128 \cdot 20 = 2560$ values; the 128 scatter points represent the mean estimate for each $\mathbf{x}$. The black line indicates perfect agreement between $b_\theta^{\text{SDE}}(\mathbf{x})$ and $\log p_\theta^{\text{ODE}}(\mathbf{x})$. We expect all points to lie below this black line for $b_\theta^{\text{SDE}}$ to be a lower bound. We find that $b_\theta^{\text{SDE}}(\mathbf{x}) \leq \log p_\theta^{\text{ODE}}(\mathbf{x})$ (up to variance error), but the optimization progresses differently depending on the prior. Gradients under the surrogate push $q_\phi(\mathbf{x})$ along the black line to increase $b_\theta^{\text{SDE}}(\mathbf{x})$ without exceeding $\log p_\theta^{\text{ODE}}(\mathbf{x})$. Optimization under the exact prior proceeds more freely, although eventually achieves higher $b_\theta^{\text{SDE}}(\mathbf{x})$ at convergence. This visualization may help explain differences in the posterior estimated with the surrogate vs. exact prior.

### 5.3 Quality of posterior estimation with Bayesian approach vs. diffusion-based approaches

A popular class of methods is diffusion-based approaches (discussed in Sec. 2.2), which attempt to sample from the posterior by incorporating measurements throughout the reverse diffusion process of the diffusion model that has been trained on images from the prior. Such approaches provide fast conditional sampling,

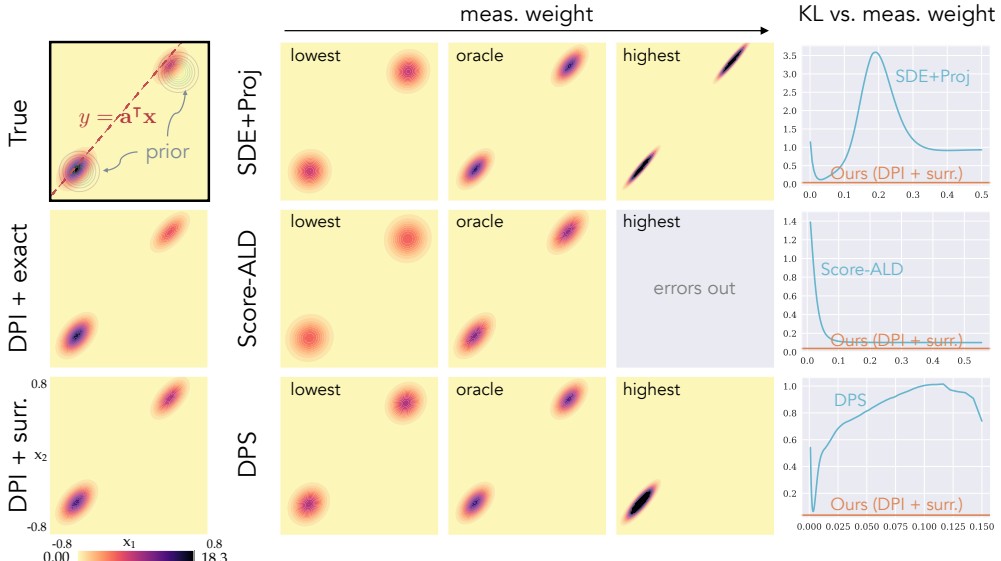

Figure 5: Comparing our VI approach with a surrogate score-based prior to baselines on a bimodal posterior. In this example, the prior is a bimodal mixture-of-Gaussians, and the likelihood is Gaussian, making the posterior a bimodal mixture-of-Gaussians (shown in "True"). Assuming access to the true prior score function, we tested how well each method recovers the true posterior. Diffusion-based methods depend on hand-tuned meas. weights. Even the meas. weight giving the best KL divergence ("oracle") does not rival using our hyperparameter-free VI approach ("DPI + surr."). Note that this "oracle" weight would not be accessible in practice, as it is determined by comparing to the ground-truth posterior. Diffusion-based baselines either (1) incorrectly place equal weight on both posterior modes or (2) miss one of the modes. DPI with either the surrogate or the exact score-based prior recovers the relative weights of both modes. **(KL vs. meas. weight)** Regardless of hyperparameters, diffusion-based methods do not reach our KL divergence.

|  | KL ($\downarrow$) | time/optimization step ($\downarrow$) |
|---|---|---|
| DPI + exact | **0.030** | 130 ms |
| Ours: DPI + surr. | 0.037 | **22 ms** |
| DPS (oracle) | 0.064 | |
| Score-ALD (oracle) | 0.10 | |
| SDE+Proj (oracle) | 0.12 | |

Table 2: Quantitative results for Fig. 5. A two-component Gaussian mixture model (GMM) was fit to estimated samples to obtain a PDF. "Ours" achieves a much lower KL div. (i.e., reverse KL from estimated posterior to true posterior) than diffusion-based baselines at their best. Time/step for DPI optimization is lower with our surrogate than with the exact score-based prior without sacrificing much accuracy.

but they may severely mischaracterize the posterior. Although our approach also approximates the posterior (due to using both a surrogate prior and a variational distribution), we find that being grounded in variational inference helps us obtain a more accurate posterior *and* images that more accurately reflect the true image than diffusion-based methods that make ad-hoc approximations of the posterior.

In the following experiments, we compare to three diffusion-based baselines: **SDE+Proj** (Song et al., 2022), **Score-ALD** (Jalal et al., 2021), and Diffusion Posterior Sampling (**DPS**) (Chung et al., 2023). SDE+Proj projects images onto a measurement subspace. Score-ALD and DPS strongly approximate the posterior throughout reverse diffusion. All baselines involve a measurement-weight hyperparameter. Our approach is DPI with the surrogate prior (i.e., using a RealNVP as the variational distribution).

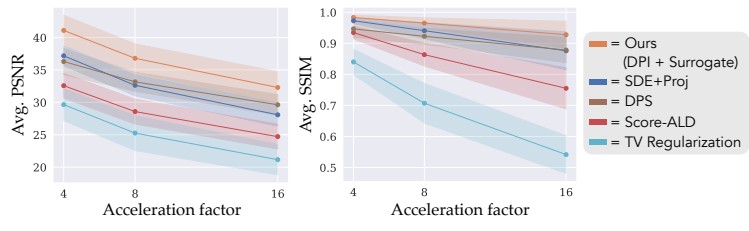

(a) Image-restoration metrics.

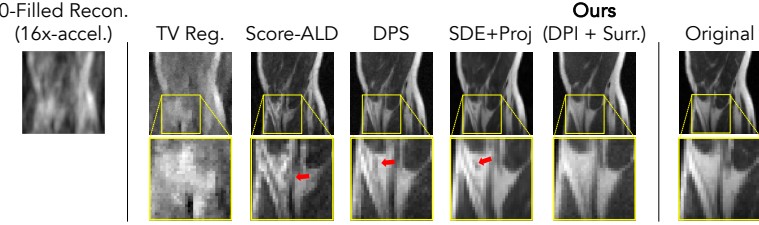

(b) Example image reconstructions for 16× acceleration.

Figure 6: Accelerated MRI. **(a)** For each accel. factor (4×, 8×, 16×), we estimated posteriors for ten knee images measured at that accel. rate. For each method, we computed the average PSNR and SSIM of 128 estimated posterior samples. The line plot shows the average result across the ten tasks; the shaded region shows one std. dev. above and below the average. **(b)** An example of 16×-accel. MRI. The cropped region exemplifies how diffusion-based baselines hallucinate more features than necessary. (a) and (b) are evidence that a principled Bayesian approach can get closer to the true image than previous unsupervised methods.

### 5.3.1 Accuracy of posterior

A simple 2D example illustrates the accuracy gap between the diffusion-based baselines and our VI approach. We consider a bimodal posterior: the prior is a bimodal mixture-of-Gaussians and the forward model a linear projection with Gaussian noise, making the posterior a bimodal mixture-of-Gaussians. This setup lets us evaluate with a true posterior and over a reasonable space of hyperparameters for baselines.

We tested how well each of the methods could recover the ground-truth posterior when given the true score function of the bimodal prior (thus avoiding potential error caused by learning the score function). Fig. 5 shows estimated probability density functions (PDFs) for the evaluated methods. None of the diffusion-based baselines correctly recover the bimodal posterior for any hyperparameter value. In particular, they struggle to find the correct balance between the two posterior modes — in the best case, they incorrectly place equal weight on each mode; in the worst case, they only recover one mode. Our VI approach with the exact or surrogate score-based prior recovers both modes in correct proportion. As shown in Fig. 5 and Tab. 2, even the best KL divergence obtained by the diffusion-based baselines does not rival that of VI with a score-based prior. We emphasize that the hyperparameter values resulting in the "best" KL for diffusion-based methods can only be found with knowledge of the ground-truth, which is inaccessible in real-world scenarios. In contrast, our method automatically finds a better KL divergence by following the Bayesian formula.

### 5.3.2 Image-reconstruction quality

It would be reasonable to assume that the diffusion-based baselines, though less principled, may lead to better visual quality than a Bayesian approach. However, we find that in addition to providing more reliable uncertainty, our approach achieves higher-fidelity reconstructions. We note that similarity to a ground-truth image does not indicate a correct posterior. Still, for a good prior, it might be desirable for posterior samples to accurately recover the original image.

We performed multiple MRI tasks at different acceleration rates and compared our approach to the diffusion-based baselines, as well as a total variation (TV) baseline. Implementations and hyperparameter settings for

SDE+Proj and Score-ALD were provided by Song et al. (2022). For DPS, we followed the implementation of Chung et al. (2023) and performed a hyperparameter search on an 8×-acceleration test image to find the optimal PSNR. For the TV baseline, we performed DPI with TV regularization with a regularization weight of $10^5$ instead of the surrogate score-based prior.

We simulated MRI at three acceleration factors for ten test images, resulting in thirty posteriors to be estimated. As baseline implementations do not assume measurement noise, we gave the baselines noiseless measurements and set a near-zero measurement noise for our method. The test images were randomly drawn from the fastMRI dataset and resized to $64 \times 64$. The score model $\mathbf{s}_\theta$ was trained on $64 \times 64$ fastMRI knee images and stayed fixed across all methods.

Our method achieves a marked improvement in PSNR and SSIM (Fig. 6). Across all acceleration factors and diffusion-based baselines, our method improves PSNR by between 2.7 and 8.5 dB. Even though our method and the diffusion-based methods all use the same score model, restoration quality depends on how the prior is used for inference; whereas baselines loosely approximate the posterior and involve hyperparameters, our approach treats the diffusion model as a standalone prior in Bayesian inference. Furthermore, Fig. 6 confirms that a diffusion-model prior far outperforms a traditional regularizer like TV.

### 5.4 Application case study: black-hole interferometric imaging

Computational imaging of distant black holes is possible with very-long-baseline interferometry (VLBI) (Thompson et al., 2017), by which a network of telescopes measures spatial frequencies of the sky's image. The Event Horizon Telescope (EHT) Collaboration used this technique to image the black hole at the center of the galaxy M87 (EHTC, 2019; 2024) and the black hole at the center of our galaxy, SgrA* (EHTC, 2022).

Imaging from EHT measurements requires a prior since telescope observations are corrupted and sparsely sample the low spatial-frequency space. Previously, the EHT Collaboration imposed hand-crafted regularizers to obtain the M87* and SgrA* images. Here we demonstrate the applicability of our method to black-hole imaging from EHT measurements. In contrast to the regularized maximum-likelihood (RML) approaches used by the EHT Collaboration (EHTC, 2019; 2024; 2022), our approach is hyperparameter-free and provides a rich data-driven posterior.

Black-hole imaging exemplifies an application that calls for a theoretically sound imaging approach. A set of telescope measurements must be carefully analyzed not only because it is expensive to collect but also because the inferred confidence intervals inform downstream scientific analysis. Therefore, an imaging approach like ours that estimates the posterior in a principled way is often desirable, even if it requires more computational resources.

#### 5.4.1 Interferometry background

In VLBI, each pair of telescopes $i, j$ provides a Fourier measurement called a *visibility* $v_{ij}$ (van Cittert, 1934; Zernike, 1938). To overcome amplitude and phase errors, we form robust closure quantities out of the measured visibilities (Blackburn et al., 2020). A *closure phase* is formed from a triplet of telescopes $i, j, k$ and robust to phase errors. A *log closure amplitude* is formed from a set of four telescopes $i, j, k, \ell$ and robust to amplitude errors. They are given respectively as

$$y_{ijk}^{\mathrm{cp}} = \angle \left( v_{ij} v_{jk} v_{ki} \right) \quad \text{and} \quad y_{ijk\ell}^{\mathrm{logca}} = \log \left( \frac{|v_{ij}||v_{k\ell}|}{|v_{ik}||v_{j\ell}|} \right). \tag{12}$$

For the inverse problem, our measurements are all the non-redundant closure phases and log closure amplitudes, which have a nonlinear forward model and Gaussian thermal noise. Fig. 7a illustrates the VLBI measurements obtained by an EHT telescope array. These particular measurements were generated from an image of a black-hole simulation. The $(u, v)$ coverage is sparse and constrained to low spatial frequencies. The "dirty" image, which is a naïve reconstruction from the measured visibilities, is completely un-interpretable. The target image is the original image blurred to the maximum resolution achievable by the EHT. Reconstructing an image beyond this intrinsic resolution is considered super-resolution.

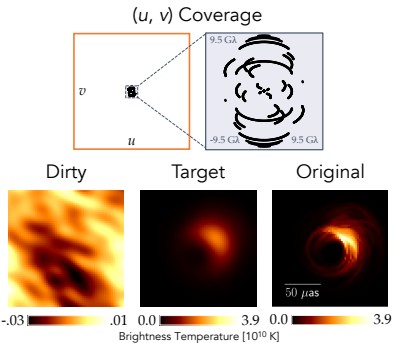

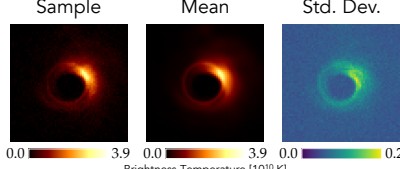

(a) Simulated VLBI measurements.

(b) Estimated image posterior.

Figure 7: Black-hole imaging. **(a)** We simulated EHT VLBI measurements with realistic noise of a synthetic black-hole image. "$(u, v)$ Coverage" shows which points in the complex 2D Fourier plane are measured by the EHT array of radio telescopes. "Dirty" is a naïve reconstruction from the sparse measured visibilities. "Target" represents the best-possible image reconstruction if the low spatial frequencies (up to the intrinsic resolution of the EHT) were to be fully and perfectly measured. "Original" is the actual underlying image that generated the simulated measurements. **(b)** Using our variational-inference approach with a score-based prior trained on simulations of black holes, we approximated the image posterior. The posterior samples recover the ring and brightness asymmetry of the original image. The pixel-wise mean and std. dev. are also shown. The std. dev. indicates regions of uncertainty, such as possible wisps inside and outside the thin ring.

### 5.4.2 Simulated-data example

We used our method to estimate an image posterior given the simulated measurements visualized in Fig. 7a. The variational distribution is a RealNVP, and the score-based prior was trained on $64 \times 64$ images of fluid-flow simulations of black holes (Wong et al., 2022). Fig. 7b shows a sample from the estimated posterior, as well as the mean and standard deviation. By using a score-based prior trained on black-hole simulations, we are able to reconstruct images that resemble these detailed simulations while still fitting the measurements.

Our proposed surrogate prior made it possible to approximate this posterior in a reasonable amount of time. Optimization with the exact prior (Feng et al., 2023) would have taken 151 GB of GPU memory and 1.8 minutes per iteration. In contrast, optimization with the surrogate prior used 46 GB of GPU memory and 177 milliseconds per iteration on the same hardware (4x 48GB NVIDIA RTX A6000).

## 6 Conclusion

We have presented a surrogate function that provides efficient access to score-based priors for variational Bayesian inference. Specifically, the evidence lower bound $b_\theta^{\mathrm{SDE}}(\mathbf{x}) \le \log p_\theta^{\mathrm{SDE}}(\mathbf{x})$ serves as a proxy for the log-prior of an image in the Bayesian log-posterior. Our experiments with variational inference show at least two orders of magnitude in runtime improvement and significant memory improvement over the ODE-based prior. We also establish that a principled approach like ours outperforms diffusion-based methods on posterior approximation and image restoration, evidence that following a traditional Bayesian approach results in more reliable image reconstructions. This advantage is crucial for applications that call for a reliable image posterior while still leveraging the expressiveness of a score-based prior.

**Broader impacts**   There are many applications, such as in the medical and astronomical domains demonstrated in this paper, in which there is incomplete information in measurements that makes priors necessary to produce interpretable images. Our work proposes a way to efficiently incorporate rich diffusion-model priors in a rigorous imaging framework. In general, however, generative models should be used with caution. They have been used to create deepfakes (Ricker et al., 2022) and may induce spurious hallucinations. Diffusion models have also been shown to memorize training data (Carlini et al., 2023; Somepalli et al., 2023), in which case one should be careful to preserve privacy when deploying a diffusion model as an image prior.

## Acknowledgments

The authors would like to thank Yang Song for his many technical insights and helpful feedback on the paper. BTF and KLB acknowledge funding from NSF Awards 2048237 and 1935980 and the Amazon AI4Science Partnership Discovery Grant. BTF is supported by the NSF GRFP.

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

# A Forward models

In this appendix, we describe the forward models of the inverse problems explored in the main text: accelerated MRI, denoising, and reconstruction from low spatial frequencies ("deblurring"). These tasks have forward models of the form

$$\mathbf{y} = \mathbf{A}\mathbf{x} + \epsilon, \quad \epsilon \sim \mathcal{N}(\mathbf{0}, \sigma_{\mathbf{y}}^2, \mathbf{I}), \tag{13}$$

with the corresponding log-likelihood function

$$\log p(\mathbf{y} \mid \mathbf{x}) \propto -\frac{1}{2\sigma_{\mathbf{y}}^2} \|\mathbf{y} - \mathbf{A}\mathbf{x}\|_2^2. \tag{14}$$

## A.1 Accelerated MRI

Accelerated MRI collects sparse spatial-frequency measurements in $\kappa$-space of an underlying anatomical image. As the acceleration rate increases, the number of measurements decreases. The forward model can be written as

$$\mathbf{y} = \mathbf{M} \odot \mathcal{F}(\mathbf{x}^*) + \epsilon, \quad \epsilon \sim \mathcal{N}(\mathbf{0}, \sigma_{\mathbf{y}}^2 \mathbf{I}), \tag{15}$$

where $\mathbf{x} \in \mathbb{C}^D$ and $\mathbf{y} \in \mathbb{C}^M$. $\mathcal{F}$ denotes the 2D Fourier transform, and $\mathbf{M} \in \{0, 1\}^D$ is a binary sampling mask that reduces the number of non-zero measurements to $M << D$. Often $\sigma_{\mathbf{y}}$ is assumed to be small (e.g., corresponding to an SNR of at least 30 dB). We use Poisson-disc sampling (Usman & Batchelor, 2009) for the sampling mask. 16×-acceleration, for example, corresponds to a sampling mask with only 1/16 nonzero elements.

**Experimental setup**  In our experiments, we assumed that $|\sigma_{\mathbf{y}}|$ is 0.05% of the DC (zero-frequency) amplitude. This corresponds to a maximum SNR of 40 dB. The only exception is for comparison to baselines (Fig. 6), since baseline methods do not account for measurement noise. In this case, we let $|\sigma_{\mathbf{y}}| = 0.1\%$ of the DC amplitude along the horizontal direction of the true image, which amounts to a very low level of noise.

## A.2 Denoising

The denoising forward model is simply

$$\mathbf{y} = \mathbf{x} + \epsilon, \quad \epsilon \sim \mathcal{N}(\mathbf{0}, \sigma_{\mathbf{y}}^2, \mathbf{I}), \tag{16}$$

where $\mathbf{x} \in \mathbb{R}^D$, and $\sigma_{\mathbf{y}}$ determines the level of i.i.d. Gaussian noise added to the clean image to get $\mathbf{y} \in \mathbb{R}^D$.

**Experimental setup**  In our presented experiments on denoising, $\sigma_{\mathbf{y}} = 0.2$, which is 20% of the dynamic range.

## A.3 Deblurring

We refer to the task of reconstruction from the lowest spatial frequencies as deblurring. The forward model is given by

$$\mathbf{y} = \mathbf{D}\mathbf{x} + \epsilon, \quad \epsilon \sim \mathcal{N}(\mathbf{0}, \sigma_{\mathbf{y}}^2, \mathbf{I}), \tag{17}$$

where $\mathbf{x} \in \mathbb{C}^D$, $\mathbf{y} \in \mathbb{C}^M$, and $\mathbf{D} \in \mathbb{C}^{M \times D}$ performs a 2D discrete Fourier transform (DFT) with only the first $M$ DFT components.

**Experimental setup**  In our presented experiments on deblurring, the measurements are the lowest 6.25% of the DFT components, and $|\sigma_{\mathbf{y}}| = 1$.

## B  Experiment details

For the sake of reproducibility, we detail the experimental setup behind each figure. Our code will be made publicly available. Some common implementation details are that the exact prior $(\log p_\theta^{\mathrm{ODE}})$ was always estimated with 16 trace estimators. The RealNVP had 32 affine-coupling layers unless stated otherwise.

### B.1  Variational distributions

We first describe the two types of variational distributions considered in our experiments: a RealNVP normalizing flow and a multivariate Gaussian with diagonal covariance.

**RealNVP**  The architecture of the RealNVP is determined by the number of affine-coupling layers and the width of each layer. For images up to $64 \times 64$, we use 32 affine-coupling layers and set the number of hidden neurons in the first layer to $1/8$ of the image dimensionality (e.g., $32 \cdot 32 \cdot 3/8$ for $32 \times 32$ RGB images). We use batch normalization in the network. Please refer to the original DPI (Sun & Bouman, 2021) PyTorch implementation[3] for details on the architecture. Our implementation is an adaptation of this codebase in JAX.

**Gaussian**  Other experiments use a multivariate Gaussian distribution with a diagonal covariance matrix as the variational family. In this case, the parameters are the mean image and the pixel-wise standard deviation. We initialize the mean at 0.5 and the standard deviation at 0.1 for all pixels. To sample, we take the absolute value of the standard deviation and construct the diagonal covariance matrix.

### B.2  MRI efficiency experiment (Fig. 1, Tab. 1)

**Score model**  For each image size, the score model was an NCSN++ architecture with 64 filters in the first layer and trained with the VP SDE with $\beta_{\min} = 0.1$, $\beta_{\max} = 10$.

**Variational optimization**  For each task (i.e., each image size and prior), the variational distribution was a multivariate Gaussian with diagonal covariance. The batch size was 64, learning rate 0.0002, and gradient clip 1. A convergence criterion based on the loss value is difficult to define due to high variance of the loss (we used 1 time sample to estimate $b_\theta(\mathbf{x})$). We defined a convergence criterion based on the change in the mean of the variational distribution. Specifically, every 10000 steps, we evaluated a snapshot of the variational Gaussian and computed $\delta = \|\mu_{\mathrm{curr}} - \mu_{\mathrm{prev}}\| / \|\mu_{\mathrm{prev}}\|$, where $\mu_{\mathrm{curr}}$ and $\mu_{\mathrm{prev}}$ are the current and previous snapshot means, respectively. If $\delta < \varepsilon$ for some threshold $\varepsilon$ two snapshots in a row, then the optimization was considered converged. Since convergence rate depends on the image size and the prior used, we set a different $\varepsilon$ for each task:

- $16 \times 16$ (surrogate): $\varepsilon = 0.002$
- $32 \times 32$ (surrogate): $\varepsilon = 0.003$
- $64 \times 64$ (surrogate): $\varepsilon = 0.005$
- $128 \times 128$ (surrogate): $\varepsilon = 0.007$
- $256 \times 256$ (surrogate): $\varepsilon = 0.009$
- $16 \times 16$ (exact): $\varepsilon = 0.0025$
- $32 \times 32$ (exact): $\varepsilon = 0.0027$
- $64 \times 64$ (exact): $\varepsilon = 0.005$

We were conservative in defining the convergence and checked that optimization under the surrogate actually achieved better sample quality than optimization under the exact prior (see Fig. 1).

---

[3] https://github.com/HeSunPU/DPI

**Data** The test image is from the fastMRI (Zbontar et al., 2018) single-coil knee test dataset and was resized to $64 \times 64$ with antialiasing.

## B.3 256x256 MRI examples (Fig. 2)

The 4×-acceleration result is from the efficiency experiment (Fig. 1 and Tab. 1) on the $256 \times 256$ test image. The 16×-acceleration result came from a similar setup, where the variational distribution was Gaussian with diagonal covariance. Optimization was done with a batch size of 64, learning rate of 0.00001, and gradient clip of 0.0002. We ran optimization for 270K steps (optimization for 4×-acceleration was done in 100K steps with the convergence criterion).

In the figure caption, we report that the true image is within three standard deviations of the inferred posterior mean for 96% and 99% of the pixels for 16×- and 4×-acceleration, respectively. This was computed based on the mean and standard deviation of 128 samples from the inferred posterior. We find the same result when using the exact mean and standard deviation of the inferred posterior: with respect to the inferred posterior, the true image is within three standard deviations of the mean for 96.7% and 99.0% of the pixels for 16×- and 4×-acceleration, respectively.

## B.4 Ground-truth posterior (Fig. 3a)

**Data** The mean and covariance of the ground-truth Gaussian prior were fit with PCA (with 256 principal components) to training data from the CelebA dataset (Liu et al., 2015). The CelebA images were resized to $16 \times 16$ with antialiasing.

**Score model** The score model was based on the DDPM++ deep continuous archictecture of Song et al. (2021b) with 128 filters in the first layer. It was trained with the VP SDE with $\beta_{\min} = 0.1$ and $\beta_{\max} = 20$ for 100K steps.

**Variational optimization** The variational distribution was a RealNVP. Under the surrogate prior, optimization was done with a learning rate of 0.00005 and gradient clip of 1. Under the exact prior, the learning rate was 0.0002 and gradient clip 1. Both priors used a batch size of 64.

## B.5 32x32 image denoising (Fig. 3b)

**Variational optimization** For both CelebA denoising (i) and CIFAR-10 denoising (ii), the variational distribution was a RealNVP. Optimization under the exact prior was done with a learning rate of 0.0002 and gradient clip of 1 for 20K steps. Optimization under the surrogate prior was done with a learning rate of 0.00001 and gradient clip of 1. For CelebA, the batch size was 64 and training was done for 1.72M steps (convergence was probably achieved earlier, but we continued training to be conservative). For CIFAR-10, the batch size was 128 and training was done for 550K steps.

**Score model** For both (i) and (ii), the score model had an NCSN++ architecture with 64 filters in the first layer. For the CelebA prior, it was trained with the VP SDE with $\beta_{\min} = 0.1$ and $\beta_{\max} = 20$ and with images that were resized without antialiasing. For the CIFAR-10 prior, it was trained with the VP SDE with $\beta_{\min} = 0.1$ and $\beta_{\max} = 10$.

**Data** The CelebA and CIFAR-10 images are both $32 \times 32$. The CelebA image was resized without antialiasing.

## B.6 Bound gap (Fig. 4)

Visualization of the bound gap is shown for optimization of the RealNVP from Fig.3b(i) (i.e., $32 \times 32$ CelebA denoising). For the plots comparing the lower-bound to the ODE log-probability, we used 2048 time samples to estimate $b_\theta(\mathbf{x})$.

### B.7 Accuracy of posterior (Fig. 5, Tab. 2)

**Variational optimization** For both the exact score-based prior and surrogate score-based prior, the variational distribution was a RealNVP with 16 affine-coupling layers, and it was optimized for 12K iterations with a batch size of 2560 and learning rate of $10^{-5}$. For the surrogate score-based prior, $b_\theta(\mathbf{x})$ was approximated with $N_t = N_z = 1$.

**Baselines** For this 2D experiment, we implemented the diffusion-based baselines exactly according to their proposed algorithms. For SDE+Proj, we tested the following values for the measurement weight $\lambda$: `linspace(0.001, 0.5, num=100)`. For Score-ALD, we distilled all hyperparameters into one global hyperparameter $1/\gamma_T$ and tested the following values for $\gamma_T$: `linspace(100, 0.8, num=100)`. For DPS, we tested the following values for the scale parameter $\zeta$: `exp(linspace(log(0.001), log(0.15), num=100))`.

**Evaluation** Since the diffusion-based approaches only provide samples (not probability densities), we approximated the probability density function (PDF) from the estimated posterior samples. For each method, we fit a two-component Gaussian mixture model (GMM) to 10000 samples. The reverse KL divergence was approximated with the log-density function of the fitted GMM and the log-density function of the true posterior, evaluated on these 10000 samples.

### B.8 Image-restoration metrics (Fig. 6)

**Score model** The score model was the same as the one used for the $64 \times 64$ image in the MRI efficiency experiment (Fig. 1).

**Variational optimization** The variational distribution was a RealNVP. Optimization was done with a learning rate of 0.00001 and gradient clip of 0.0002. We used the same convergence criterion as the one used in the MRI efficiency experiment with $\varepsilon = 0.005$. The same convergence criterion was also used for the TV results but with a maximum number of steps of 50000. The TV regularization weight was $10^5$.

**Baseline hyperparameters** For SDE+Proj, we used the `projection` CS solver provided by Song et al. (2022) with the hyperparameters `snr=0.517, coeff=1`. For Score-ALD, we used the `langevin` CS solver with the hyperparameters `n_steps_each=3, snr=0.212, projection_sigma_rate=0.713`. For DPS, we used `scale=0.5`. This was the best scale out of $[10, 1, 0.9, 0.5, 0.3, 0.1, 0.001]$ for a test image in terms of PSNR with respect to the true image.

### B.9 Black-hole interferometric imaging (Fig. 7)

**Score model** The score model was trained on images of general relativistic magneto-hydrodynamic (GRMHD) simulations (Wong et al., 2022) resized to $64 \times 64$. During training, the images were randomly flipped horizontally, and they were randomly zoomed so that the ring diameter would vary between 35 and 48 $\mu$as. The score model had an NCSN++ architecture with 64 filters in the first layer; it was trained with the VP SDE with $\beta_{\min} = 0.1$ and $\beta_{\max} = 20$ for 100K steps.

**Variational optimization** The variational distribution with a RealNVP. Optimization was done with a learning rate of 0.00001 and gradient clip of 1 for 100K steps.

**Interferometric imaging assumptions** Imaging was done for a field of view of 160 $\mu$as. A flux-constraint loss was added to the DPI optimization objective so that all posterior images would have a total flux around 173 (computed as the median total flux of images sampled from the score-based GRMHD prior).

