# OpenReview forum: "Variational Bayesian Imaging with an Efficient Surrogate Score-based Prior"
_TMLR — Accepted by TMLR_

### Review · Reviewer_jZTo · 2024-04-11

**Summary Of Contributions:**

The paper proposes a variational inference based surrogate modeling procedure for diffusion models that can characterize complicated distributions by use of score based priors. The approach is found to accelerate inference by orders of magnitudes when compared to exact ODE based prior evaluations. The approach is considered in the context of ill-posed image reconstruction in which the validity of the prior imposed can have substantial influence on the quality of image reconstruction. On synthetic and real datasets the merits of the proposed approach is established when compared to exact modeling and some existing diffusion based baselines.

The approach is leveraging the work of (Feng et al., 2023) establishing how score based diffusion models can be used as priors relying on expensive ODE solvers thus the need for an efficient surrogate modeling approach scaling this procedure. The approach here explores that the evidence lower bound of the ODE based prior can be efficiently computed leveraging the work of (Song et al., 2021a) to define such efficient ELBO based surrogate function optimized using importance sampling and MC approximations.

**Audience:**

Yes

**Broader Impact Concerns:**

Inverse models are influenced by the inductive biases imposed through the prior and it would therefore be good to further elaborate on the trustworthiness of such reconstructions and if reconstructions can be hallucinated resulting in critical erroneous interpretations. Furthermore, diffusion models can negative influence society in terms of deep fake technologies and disinformation – could the considered technology contribute to such negative aspects?

It would be good to include discussions of the broader impact of the considered research, which is presently lacking from the current manuscript.

**Claims And Evidence:**

Yes

**Requested Changes:**

Why have traditional priors as reviewed end section 2.1 not been compared against in the experimentation and assessment of results?

It is well known that ELBOs in general can be tightened by importance weighting. Could such importance weighted ELBOs improve upon the surrogate function accuracy and would this be useful?

There are many strategies for speeding up the computations associated with diffusion models and the paper has not positioned the current work in this larger literature. It would improve the paper to do this and also explain why some of these other efforts are not relevant/useful in the considered context, i.e. latent diffusion, reducing steps of the diffusion, reverse processes learned to be reduced in diffusion steps etc. Would these existing approaches be relevant also in the considered context?

Please include a broader impact discussion (see also next section)

**Strengths And Weaknesses:**

Strengths

•	The proposed approach is useful and provides an accurate surrogate function with orders of magnitude improved efficiency providing a useful computational framework using score-based diffusion models as priors.

•	The approach is sound.

•	The results are compelling.

Weaknesses

•	The approach technically combines insights from (Feng et al., 2023) with estimation from (Song et al., 2021a). As such, the novelty lies in combining score-based diffusion priors with efficient ELBO based surrogate functions for the problem of ill-posed image reconstruction. The novelty from a technical perspective is thus somewhat incremental.

•	The approach could be contrasted also conventional image reconstruction procedures based on standard priors as reviewed at the end of section 2.1. It is unclear how the approach compares to non-diffusion based procedures.

•	The paper includes no broader impact discussion which should be included.

---

### Review · Reviewer_qKCT · 2024-04-22

**Summary Of Contributions:**

This paper proposes an approximate inference technique for Bayesian imaging where the prior is defined by a pretrained score-based diffusion model.
Specifically, the authors use the ELBO as a substitute for the exact log-probability function and show, empirically, that compared to the exact approach of Feng et al. (2023) this yields large computational savings while resulting in a close approximation of the posterior.

**Audience:**

Yes

**Broader Impact Concerns:**

No.

**Claims And Evidence:**

Yes

**Requested Changes:**

I don't have requests I consider critical, but please consider my suggestions above to strengthen the work.

**Strengths And Weaknesses:**

**Strengths:**

- The paper is clear, well-written, and logically presented.

- Experiments are clearly defined and answer specific questions.

- The empirical results show dramatic improvements in computational efficiency.

- The code will be made publically available (according to the appendix).

**Weaknesses:**

- While the theory is clearly presented, it is mostly a recap of the methods developed in Song 2021a. A theoretical analysis of, for instance, how the approximation error in the learned score function affects performance would have strengthened the paper.

**Comments and suggestions:**

The setting considered is where an unconditional diffusion model is trained a priori. For this to make sense, a dataset of sufficiently related images needs to exist. A discussion of where this situation arises could be helpful, which could be contrasted to blind image restoration where paired (high quality-low quality) images are available. Furthermore, it would have been interesting to understand the effect of domain shifts between the data the prior is trained on and that of the actual measurements. That said, I understand that the focus of this paper is on the quality of the approximate inference and not on comparing different (learned) priors.

In section 5.2, does the prior have a significant influence? My suggestion would be to add results from a maximum likelihood approach or a simple conventional prior like TV or L1.


**Nitpicking:**
 - Rephrase "In an ill-posed inverse problem, there is inherent uncertainty in image reconstruction."
 - more-accurate -> more accurate
 - Avoid starting sentences with mathematical expressions.

---

### Review · Reviewer_7FR5 · 2024-05-03

**Summary Of Contributions:**

The paper studies the use of a score based diffusion model as a prior in Bayesian imaging. The exact computation of probability densities under the diffusion model is expensive, and the paper uses an approximation that is orders of magnitude faster. This is demonstrated on several examples (simulated as well as real) and validated in terms of efficiency, posterior estimation, and reconstruction quality.

**Audience:**

Yes

**Claims And Evidence:**

Yes

**Requested Changes:**

Please revise the abstract so that the problem setting is more clearly defined.

Can you more clearly define from the onset what is technically meant by "ill-posed image reconstruction". Perhaps just a short description along the lines of the first few sentences in sec. 2.1, so the setting is clear from the onset.

Is there a missing "dt" in equation 2?

Could you provide references for eq. 2 and 3. (perhaps Song 2021b or Anderson 1982). Would it be possible to also provide some intuition about eq. 3 or some hint to how it is derived?

In section 4: "Inspired by previous ... we propose bSDE ..." Consider rephrasing 'propose' to 'apply' or similar, to clarify that bSDE is not a novel contribution.

I am not so familiar with the imaging application, so I am unsure if the proposed method is evaluated fairly against appropriate baselines.

**Strengths And Weaknesses:**

Strengths.
The use of diffusion models as strong priors is interesting and innovative. The argumentation for modeling choices is clear and  coherent.
The paper provides a fairly clear and concise technical summary of the score based diffusion framework.
Writing is clear and concise.
The practical relevance and potential impact is strong.

Weaknesses.
Beacause I am not so familiar with the field, it is unclear to me if the comparison with other approaches is fair and comprehensive.
I did not see a link to a software implementation. I would strongly suggest providing code.

---

### Author Response · Authors · 2024-05-31
**Rebuttal**

We thank all the reviewers for their positive feedback and constructive suggestions. We are glad that they appreciated the “practical relevance and strong impact” (7FR5), “compelling” empirical results (qKCT, jZTo), and clear writing (7FR5, qKCT) of our manuscript. We have submitted a rebuttal with the requested revisions and address general concerns here.

**Baseline comparisons (7FR5, qKCT, jZTo).** We included the same diffusion-based baselines as those in Feng et al. (2023), which are well-regarded methods in the field. We kept the comparisons fair: for SDE+Proj and Score-ALD, we used the implementations and hyperparameters of Song et al. (2022); for DPS, we simply translated the implementation of Chung et al. (2023) from PyTorch to JAX. As mentioned in Appendix B.8, we used the best DPS measurement weight (out of seven tested values) in terms of PSNR. It would be infeasible to cover all of the different diffusion-based inverse solvers proposed in recent years, so we chose these three baselines that have different ways of approximating the posterior. DPS is regarded as the most popular of the diffusion-based approaches, garnering 285 citations from the time of its publication in 2022 until now. As for comparisons to a traditional prior, we have added TV results to Figure 6.

**Technical (jZTo) and theoretical (qKCt) contributions.** Our main contribution comes from connecting the insights of Song et al. (2021a) with traditional variational Bayesian imaging. The usefulness of the bound derived by Song et al. for solving inverse problems in a Bayesian formulation is only trivial once this connection is made. We agree that theoretical analysis would further our understanding of the expected gap between the estimated posterior and the true posterior. However, the focus of our work is to demonstrate the practical usefulness of the surrogate function for imaging. Thus our work brings “practical relevance and strong impact” (7FR5), as validated through “compelling” empirical results (qKCT, jZTo).

**Impact of domain shifts between training data and actual measurements (qKCT).** Feng et al. (2023) performed a thorough analysis of prior mismatch in their Figures 7, 8, and 9. They found that more mismatch between the prior and actual measurements causes more error in the reconstruction, as expected. However, they also found that the Bayesian variational approach leads to posterior reconstructions that are more stable in spite of prior mismatch than the diffusion-based baselines (i.e., Score-ALD, SDE+Proj, and DPS). Although selecting a suitable dataset for the prior is an important topic, it is not the focus of our work. Instead, we address the problem of how to incorporate the prior in an efficient way regardless of the training dataset.

**Importance weighted ELBO (jZTo).** We thank jZTo for the idea of improving the proposed surrogate function with importance weighting. Our work focuses on the ELBO as originally proposed by Song et al. (2021a) in order to first establish its utility as a prior for inverse imaging.

**Codebase (7FR5).** The footnote on page 2 clarifies that we will add the link to the codebase upon publication. There is already a public codebase for this work, but we did not include the link to preserve anonymity during the review process.

---

### Decision · Action_Editor_3RDN · 2024-06-24

**Recommendation:** Accept as is

**Comment:**

In this paper, the authors focus on inverse problem with deep priors given by diffusion models. Here they train a model using the forward KL to match the posterior in a setting where the likelihood is known and the log probability of the prior is lower bounded with a continuous time ELBO. The authors successfully apply their methods to accelerated MRI, denoising, reconstruction from low spatial frequencies, and black-hole interferometric imaging. The methodology is sound. The experiments are quite compelling. Based on this assessment and the feedback of the reviewers I recommend the acceptance of the manuscript as is.

**Audience:**

Yes.

**Claims And Evidence:**

Yes.